# Real-World Data on EGFR and ALK Testing and TKI Usage in Norway—A Nation-Wide Population Study

**DOI:** 10.3390/cancers15051505

**Published:** 2023-02-27

**Authors:** Inger Johanne Zwicky Eide, Yngvar Nilssen, Elin Marie Stensland, Odd Terje Brustugun

**Affiliations:** 1Section of Oncology, Drammen Hospital, Vestre Viken Hospital Trust, N-3004 Drammen, Norway; 2Institute of Clinical Medicine, University of Oslo, N-0316 Oslo, Norway; 3Cancer Registry of Norway, N-0304 Oslo, Norway

**Keywords:** EGFR, ALK, non-small cell lung cancer, registry, survival, population-based study

## Abstract

**Simple Summary:**

Drugs acting against specific molecular alterations in cancer cells are tested in clinical studies and found to be highly efficacious in some patients with lung cancer. However, the true benefit in patients treated outside of clinical trials is, to some extent, unknown. It is also not well-described how many of the patients that could benefit from such treatment are offered these drugs. One could imagine that the molecular alterations are not tested for, or that not all relevant patients are treated. Based on national registries in Norway, we have here analysed these issues, and found that a high fraction of patients are tested according to guidelines. Additionally, at high ages, most patients are provided therapy, and the benefit seems to replicate what is seen in the clinical trials. Thus, we may trust the study results and have confidence that the new drugs do prolong the lives of the relevant patients.

**Abstract:**

Clinical studies have shown the efficacy of EGFR- and ALK-directed therapies in non-small cell lung cancer (NSCLC). Real-world data on, e.g., testing patterns, uptake, and duration of treatment are scarce. Reflex EGFR and ALK testing of non-squamous NSCLCs were implemented in Norwegian guidelines in 2010 and 2013, respectively. We present a complete national registry data on incidence, pathology procedures, and drug prescription in the period of 2013 to 2020. Test rates for both EGFR and ALK increased over time and were 85% and 89%, respectively, at the end of the study period, independent of age up to 85 years. The positivity rate for EGFR was higher among females and young patients, whereas no sex difference was observed for ALK. EGFR-treated patients were older than ALK-treated patients (71 vs. 63 years at start, *p* < 0.001). Male ALK-treated patients were significantly younger than females at the start of treatment (58 vs. 65 years, *p* = 0.019). The time from the first dispensation to the last dispensation of TKI (as a surrogate for progression-free survival) was shorter for EGFR- than for ALK-TKI, and survival for both EGFR- and ALK-positive patients was substantially longer than for non-mutated patients. We found a high adherence to molecular testing guidelines, good concordance of mutation positivity and treatment, and the real-world replication of findings in clinical trials, indicating that the relevant patients are provided substantially life-prolonging therapy.

## 1. Introduction

Lung cancer is the deadliest of cancers and is still frequently diagnosed in incurable stages [1,2]. Most patients with non-small cell lung carcinoma (NSCLC) are classified as non-squamous variants. Of these variants, adenocarcinoma, which comprises approximately 50% of all lung cancers, is the most frequent histological subtype in both sexes [3]. Therapeutic possibilities for NSCLCs include immunotherapy with or without chemotherapy or drugs targeting specific molecular aberrations (typically tyrosine kinase inhibitors—TKIs), if present [4]. Such molecular targets are virtually exclusively found in adenocarcinomas. Targeted therapy for metastatic lung cancer has been in routine use since pivotal studies showed high response rates and durable responses in individuals with tumours harbouring specific mutations, first related to aberrations in the EGFR and ALK genes [5,6]. The molecular testing of adenocarcinomas for EGFR and ALK has been implemented in many countries, and updated international guidelines are published [7]. Although most guidelines recommend routine testing, the actual frequency of tests in eligible patients is variable [8,9,10,11]. The uptake of treatment with targeting drugs in patients where relevant targets are present is likely also variable [9,12]. The uptake of test activity may increase as a function of time since time of implementation of recommendation but may also vary due to other factors, such as the as patient’s age and sex. The efficacy of the drugs in a real-world setting may differ from published prospective study results and may also vary based on patient’s age.

Norwegian guidelines have recommended routine EGFR and ALK testing since 2010 and 2013, respectively, for all patients diagnosed with non-squamous cell carcinoma NSCLCs, irrespective of age and stage. Drugs targeting these aberrations have been approved and are generally reimbursed from the same time.

In this study, we aimed to describe Norwegian test routines and patterns of use of ALK- and EGFR-TKIs on a national level as a measure of estimating guideline adherence and treatment benefit. Specifically, we wanted to study the EGFR and ALK test frequency and positivity rate related to time period and patient age and sex. Furthermore, we wanted to study the usage of drugs targeting EGFR and ALK and correlate the duration of use and overall survival to patient characteristics.

## 2. Material and Methods

### 2.1. Data Collection

Information on histology subtypes, EGFR test activity, and ALK test activity was obtained from the Cancer Registry of Norway (CRN). EGFR testing involves different DNA-based tests, whereas ALK is assessed using a immunohistochemistry (IHC) staining or a FISH analysis. A strong (3+) IHC stain or a FISH positivity index > 15% was regarded positive for ALK. Since 1953, all newly diagnosed, malignant neoplasms have been reported to the CRN as it has been mandated by law to submit copies of cytology, biopsy, and autopsy reports from pathology laboratories as well as clinical diagnoses from hospitals. The completeness of the CRN is considered to be very high (98%) [13]. 

Information on the use of ALK- and EGFR-TKIs was retrieved from the Norwegian Prescription Database (NorPD). The NorPD was established in 2004 and records all drugs ordered and prescribed from pharmacies in Norway [14]. The database is considered reliable, updated, and valid [15]. We collected data from 1 January 2013 to 31 December 2020 for the relevant ALK- and EGFR-TKIs with ATC codes L01XE02 (gefitinib), L01XE03 (erlotinib), L01XE13 (afatinib), L01XE16 (crizotinib), L01XE28 (ceritinib), L01XE35 (osimertinib), L01XE36 (alectinib), L01XE43 (brigatinib), L01XE44 (lorlatinib), and L01XE47 (dacomitinib).

An individual is defined as a new individual (new user) at the individual’s very first delivery of ALK- or EGFR-TKI, respectively, in the period from 1 January 2013 to 31 December 2020.

The data were broken down by ALK inhibitors (ATC codes L01XE16, L01XE28, L01XE36, L01XE43, and L01XE44) and EGFR inhibitors (ATC codes L01XE02, L01XE03, L01XE13, L01XE35, and L01XE47). Age was calculated at the end of the dispensing year. For new individuals, age was calculated at the end of the first year of dispensation. The data material are based on prescriptions for individuals registered in the pharmacy with a valid social security number.

The treatment time per individual was defined as the difference between the date of the first and last dispensation of an ALK- or EGFR-TKI in the period from 1 January 2013 to 31 December 2020. The number of packages per individual was defined as the sum of the number of packages the individual received from the first to the last dispensation in the period of 1 January 2013 to 31 December 2020. Each package comprised tablets for one month’s treatment.

### 2.2. Statistical Methods

Linear regressions in which the proportion of diagnosed patients who were tested for EGFR/ALK mutations was the dependent variable and the year of diagnosis was the independent variable were performed to determine if there were statistically significant increases in the proportions over time. To test for differences between age groups, two Poisson regressions were performed in which the dichotomous variable for the mutation test (EGFR/ALK) was the dependent variable and the age group as the independent variable. In these analyses, the youngest age group (0–44 years) was considered the reference. Additional Poisson regressions in which sex was the independent variable were performed to explore sex differences in the proportion of positive mutation (EGFR/ALK) tests for (1) all ages combined, (2) ages of less than 60–64 years, and (3) ages greater than 60–64 years. *t*-tests examined age distribution differences between sexes among new users of EGFR- and ALK-TKIs separately and combined. The number of dispensed packages was converted to months as one package consistently represented one month of drug consumption, and the months were presented as medians with the corresponding interquartile intervals. A *p*-value < 0.05 was considered significant. The Kaplan–Meier approach was used to estimate the overall survival. We had vital information on all patients up until 30 June 2022. The statistical program Stata 17.0 (StataCorp. 2021. Stata Statistical Software: Release 17. College Station, TX, USA: StataCorp LLC) was used for all analyses.

## 3. Results

### 3.1. Molecular Testing

During the period under study (2013–2020), 12,532 patients were diagnosed with non-small cell lung cancer of the non-squamous subtype in Norway. EGFR- and ALK-testing was performed on 9507 and 8770 patients, respectively. Nation-wide implementation of EGFR-testing was implemented in 2010. The test rate (percentage tested of patients diagnosed with non-squamous cell carcinoma NSCLC) for EGFR has increased significantly (*p* < 0.001) over the analysed time period and has reached a plateau in recent years, being 59% in 2013 and 85% in 2020 (Figure 1). ALK testing with immunohistochemistry was recommended nation-wide in October 2013. The test rate was lifted from 14% in 2013 to 61% in 2014, and it further increased to 89% in 2020 (Figure 1). The fraction of positive test results among those tested has increased for EGFR (from 8.4 to 10.3% in 2013 and 2020, respectively, and has decreased for ALK in the same period, from 7.8% to 2.1%. 

The test rate for both EGFR and ALK in 2016–2020 was similar regardless of age except for the highest age group (85 years or older), which was approximately 85% and dropped to 60–70% in the oldest group (*p*-value: 0.007 (85+ years vs 0–44 years)) (Figure 2).

The fraction of positive tests differed between age groups and sexes. For EGFR, we found a high positivity rate in the youngest age groups (35% of females and 27% of males aged 0–44 years), and the positivity rate increased for females in the older age groups, whereas for males no such trend was evident (Figure 3). For ALK-positives, a higher positivity rate was found in younger age groups (15% and 21% for females and males, respectively, aged 0–44 years, and 11% for both sexes aged 45–49 years), whereas the positivity rate was low for both sexes in older ages (Figure 3). In the period 2013–2020, 69% of the positive EGFR tests and 57% of the positive ALK tests were in females.

### 3.2. Sex and Age of EGFR- and ALK-Treated Patients

For new users of EGFR-TKIs (total *n* = 865 in the period 2013–2020), the fraction of females was 63% for the total period. There was a non-significant trend for a higher fraction of females over time (R^2^ = 0.44, *p* = 0.074). 

In total, 55% of new ALK-TKI users (*n* = 215 in the period 2013–2020) were females. The sex distribution fluctuated over time, and no significant time trend was seen (*p* = 0.41). 

New users of EGFR-TKI are, in general, older than new users of ALK-TKIs. In the period 2016–2020, the median age were 71 years vs. 63 years (*p* < 0.001) for new EGFR- and ALK-users, respectively (Figure 4). The median age for new EGFR-TKI users was 69 years and 72 years for males and females, respectively (non-significant, *p* = 1.0). The median age for new ALK-TKI-users was 58 years and 65 years for males and females, respectively; this difference was significant (*p* = 0.019). A higher fraction of new users of ALK-TKI were younger than 50 years when compared with EGFR-TKI users (22.4% vs. 8.8%), whereas a lower percentage of new ALK-TKI users were 75 years or older (12.1% vs. 34.8%).

We also compared the number of positive tests, regardless of stage, to the number of individuals starting on EGFR- and ALK-TKI, respectively. Patients could be starting a first therapy after a diagnosis of stage IV, or at relapse after an initial earlier stage. A higher number of EGFR-TKI-treated patients than EGFR-mutation positive individuals was seen during the first part of the period. This declined to a ratio of 0.59 in 2020. For ALK, the ratio of treated patients versus ALK-IHC/-FISH-positives increased from 0.32 in 2014 (the first full year of recommended reflex-testing), to 0.70 in 2020.

### 3.3. Time on Treatment and Survival

The time from the first to the last dispensation of TKI was shorter for patients on EGFR-TKI than on ALK-TKI (Figure 5). For the period 2016–2020, the median duration for EGFR-TKI usage was 152 days (99 for males and 203 for females), whereas the median duration of ALK-TKI treatment was 229 days (279 days for males and 223 for females). When excluding EGFR-treated patients (25% of males and 10% of females) to whom treatment was dispensed only once (registered time on treatment = 0 days), the duration on treatment was 189 and 241 days for males and females, respectively. For ALK-treated patients, 5% of males and 14% of females were dispensed treatment only once, and the time on treatment remained virtually unchanged. 

We also analysed the duration of treatment based on the number of dispensed drug packages (one package equals one month on treatment) in various age groups (Figure 6). 

The median time on treatment for EGFR-TKI-treated patients was the longest in the younger age groups, with a median of 17 (IQR 4–21) months among patients aged 40–44 years. The shortest time on treatment was found in the 60–64 year age group (a median of 5 months, IQR 2–14), whereas the median time on treatment was 7 (IQR 3–16) months in patients aged 75 years or older.

For ALK-TKI, the median time on treatment was the longest in the middle age groups (45–59 years). Patients aged 50–54 years were on treatment for a median of 18 (IQR 6–33) months in the last time period (2016–2020). The median time on treatment was 5 (IQR 3–11) months in the oldest group (75+ years). 

Finally, we analysed the overall survival for patients diagnosed with non-squamous NSCLC in stage IV and correlated this to the mutational status and year of diagnosis (either 2013–2016 or 2017–2020) (Figure 7). We found a substantially better two-year survival for ALK- and EGFR-mutated patients compared to the non-mutated patients. For ALK-mutated patients, no obvious difference in survival was found depending on the period of diagnosis, whereas for EGFR-mutated patients a separation of the curves was evident after approximately 18 months of follow-up, favouring patients diagnosed in the latter period. For non-mutated patients, the two-year survival increased from approximately 10% in the first period to approximately 20% in the last.

## 4. Discussion

We have presented data on testing and treatment patterns of EGFR- and ALK-mutated lung cancer on a national level. Due to high quality registers of incidence, pathology, procedures, and prescriptions, we can investigate real-world data for large patient groups nation-wide. Furthermore, Norway has updated national guidelines on the work-up and treatment of lung cancer [16]. Therefore, we aimed to correlate real-world data to the guidelines.

We found a high adherence to testing guidelines as the number of tests per year for EGFR and ALK aberrations approached the incidence of non-squamous NSCLC. Other groups that recently published test rates found a similar or slightly lower adherence [10,11]. It is interesting to note that the test frequency was stable for all age groups up to 84 years, whereas the frequency dropped (but still remained above 70 and 60%) for EGFR and ALK, respectively, in patients aged 85 years or older. This underscores a loyalty to the implementation of routine testing of all non-squamous NSCLC that was put into action in 2010 and 2013 for EGFR and ALK, respectively.

The rates of positive EGFR tests were in the range of what is published elsewhere, although the positivity rate for ALK (around 2%) was somewhat lower than what was published by other groups [17,18]. This may partly due to the fact that our study included patients in early stage, at which ALK is less frequent than in advanced cases [19]. 

EGFR positivity was more frequent in females, especially in the higher age groups, above 70 years, whereas a higher positivity rate for both sexes was evident in the youngest age groups. For ALK, there was no difference among sexes, and the positivity rate was high in the younger age groups and remained low above 55 years of age. Interestingly, more than one third of EGFR-TKI-treated patients were 75 years or older at the time of beginning to use TKI and had a median time on treatment that paralleled younger age groups. This underscores the fact that EGFR-testing should not be omitted in the higher age groups as older patients will also tolerate EGFR-TKI-treatment, which may provide substantial life prolongation [20].

The time on ALK-TKIs was found to be similar among sexes but was markedly shorter in the youngest and oldest age groups. Both these aspects contrast the EGFR group, in which females had a longer period on treatment and there was no apparent correlation to age. It is interesting to note that a possible age effect was also found in phase III-trials such as the Profile 1014-study, in which the hazard ratio for crizotinib versus chemotherapy was 0.38 in patients 65 years or older and 0.99 in patients younger than 65 years of age [21]. It is important to note that our data did not contain survival data for ALK-treated patients specifically, and that time spent on treatment is different from overall survival. Some real-world studies have reported survival figures for ALK-treated patients in the range of 2–4 years [22,23,24], whereas a study from USA presented a summed PFS of approximately 15 months, which is in line with our data [25]. Our survival data for ALK-positive, stage IV-patients showed that more than 50% were alive after 2 years, consistent with other published real-world studies. Of note, our data included ALK-positive patients regardless of treatment, as ALK tests are performed reflexively by the pathologist without knowing the performance status of the patient.

All ALK inhibitors were solely been indicated for patients with ALK-positive tumours (defined by a validated method), whereas EGFR inhibitors (specifically erlotinib) were also used to some extent outside of EGFR-positive patients [26]. This was especially true in the first years of our study period before the clinical community became aware of the low probability of effect in EGFR-negative patients, and before the advent of other treatment options as immunotherapy. These considerations may be the reason for the reduction in use of EGFR-TKI while, at the same time, an increase in the duration of time on treatment was observed. We also observed that up to one fourth of male patients were dispensed EGFR-TKI only once, indicating that a relatively high fraction of these patients did not respond durably to treatment. This may also relate to the usage of these drugs without a positive biomarker. Why this is especially evident among males is not clear, but may be partly due to a higher chance of positivity in non-tested females (approximately 50% of Caucasian females who have never smoked harbour an EGFR-mutation) [27,28]. In our period of study, first- or second-generation EGFR inhibitors were eventually used with osimertinib as the next-line treatment, whereas osimertinib is now the preferred first-line EGFR-TKI with no obvious effective second line EGFR inhibitor [29]. For ALK-positive patients, the sequencing of various ALK inhibitors, depending on availability, is now implemented in real-world settings [30].

Our study has several limitations: the time on treatment may be underestimated due to patients remaining on treatment at the end of the study period. Patients who were treated in clinical studies and dispensed study medication are not registered in the prescription data base. Compliance is unsecure; hence, the number of packages may not reflect the true time on treatment, and time on treatment for patients with only one dispensing is, to some extent, also unknown. Furthermore, patients that would not be eligible for inclusion in clinical trials due to worse performance status, comorbidities, and other factors were likely treated in our dataset. All these factors may contribute to the shorter time on treatment than what is presented as progression-free survival in clinical trials. We present no progression-free survival data, but time on treatment is suggested, also by others, to be a relevant endpoint in TKI-treated patients [31]. We present survival data for patients diagnosed with non-squamous NSCLC in stage IV and with ALK or EGFR mutations; however, some of these NSCLCs might not have been treated with relevant targeted therapy. Nevertheless, we found a substantial survival difference between mutation-positive patients and patients without ALK- or EGFR-mutations, underscoring the life-prolonging effect of targeted drugs. Furthermore, we have no information on reasons for discontinuing treatment, which may include a toxicity known to be more pronounced in higher age groups [32]. We also have no data on specific subtypes of mutations which might be relevant with respect to response Interestingly, a higher fraction of uncommon, less-responsive EGFR- mutations have, in some studies, been found in patients younger than 50 years [33].

## 5. Conclusions

To conclude, we have in this real-world study of non-squamous NSCLC patients found a high EGFR and ALK test rate in older age groups and positivity rates as expected. Our indirect data, using time on treatment as well as overall survival for patients with mutations regardless of treatment, indicate that targeted therapies are effective across age and sex and demonstrate a substantial life-prolonging benefit.

## Figures and Tables

**Figure 1 cancers-15-01505-f001:**
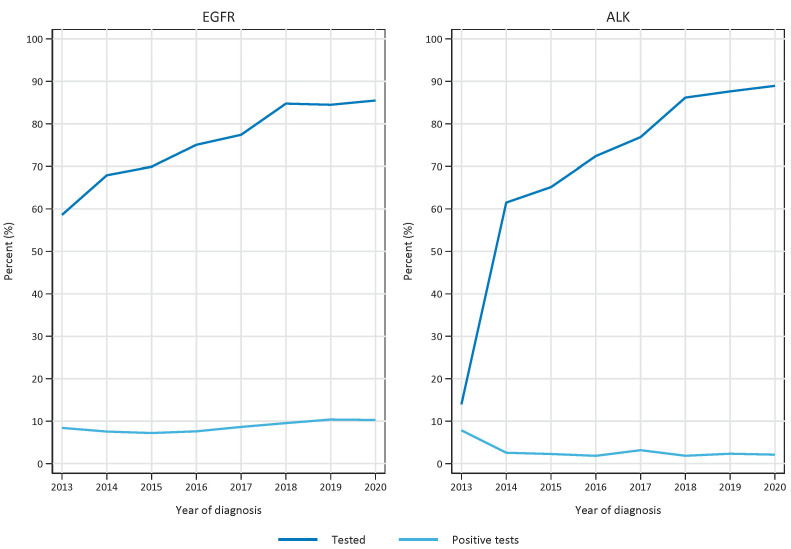
Test rate (number of tests versus number of diagnosed non-squamous non-small cell lung cancer patients) and fraction of positive tests per year in the period 2013–2020. Left panel represents EGFR tests and right panel represents ALK tests.

**Figure 2 cancers-15-01505-f002:**
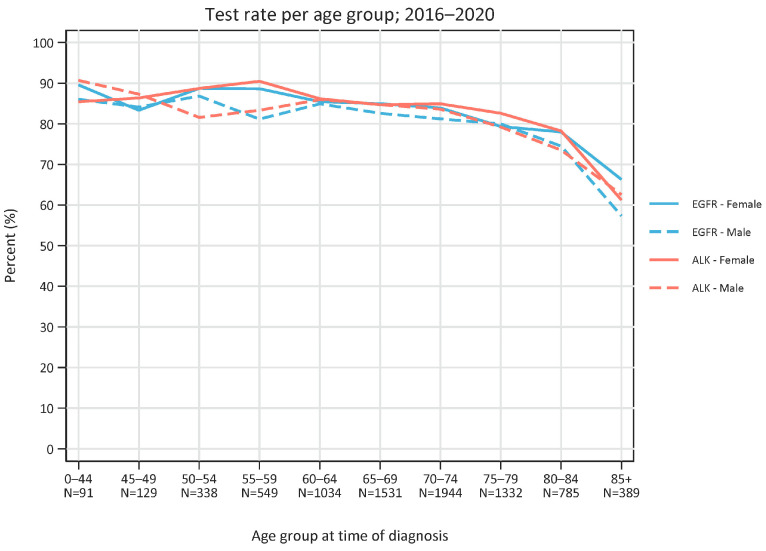
Test rate (number of tests versus number of diagnosed non-squamous non-small cell lung cancer patients) per age group in the period 2013–2020. Total number of patients in each age group is depicted. Blue lines represent EGFR tests and red lines represent ALK tests. Solid lines represent females and dotted lines represent males.

**Figure 3 cancers-15-01505-f003:**
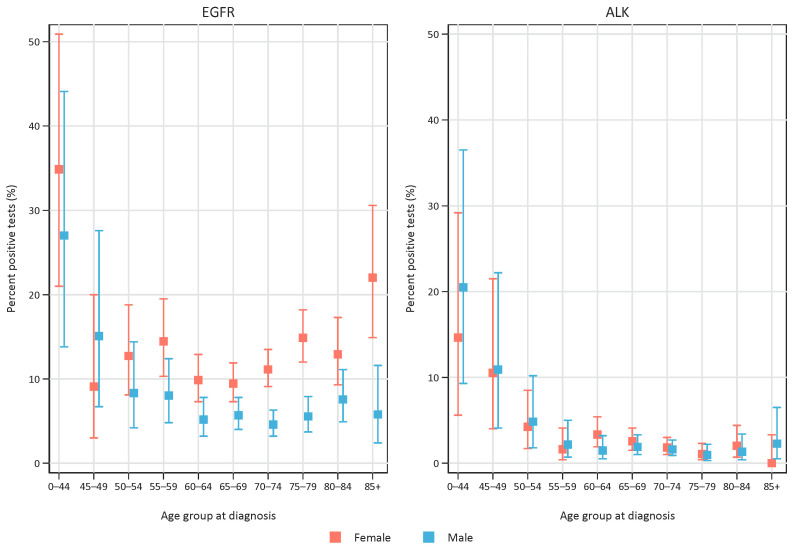
Fraction of positive test results among tested patients, related to age groups. Left panel represents EGFR tests and right panel represents ALK tests. Red symbols represent females and blue symbols represent males. Error bars represent 95% confidence interval.

**Figure 4 cancers-15-01505-f004:**
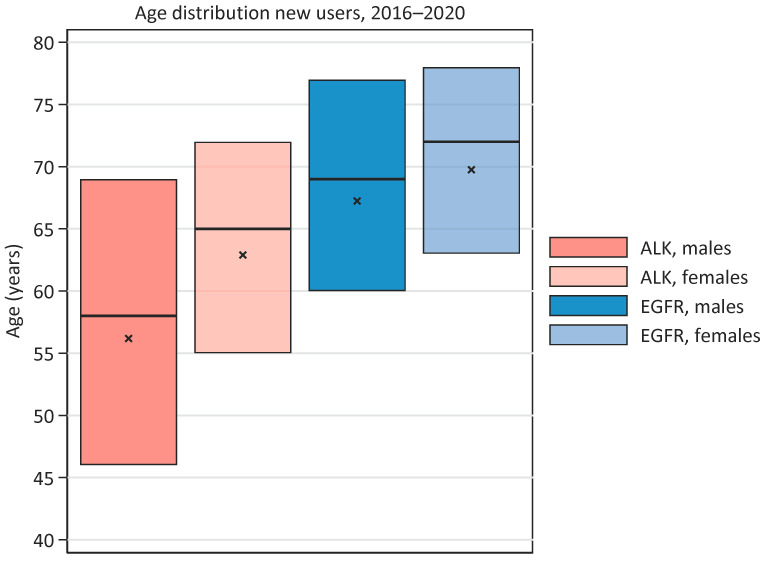
Age distribution among males and females at start of EGFR- or ALK-TKI treatment, respectively, in the period 2016–2020. Median is shown with horizontal line, mean with a cross, and range of bar represents 25- and 75-quartiles.

**Figure 5 cancers-15-01505-f005:**
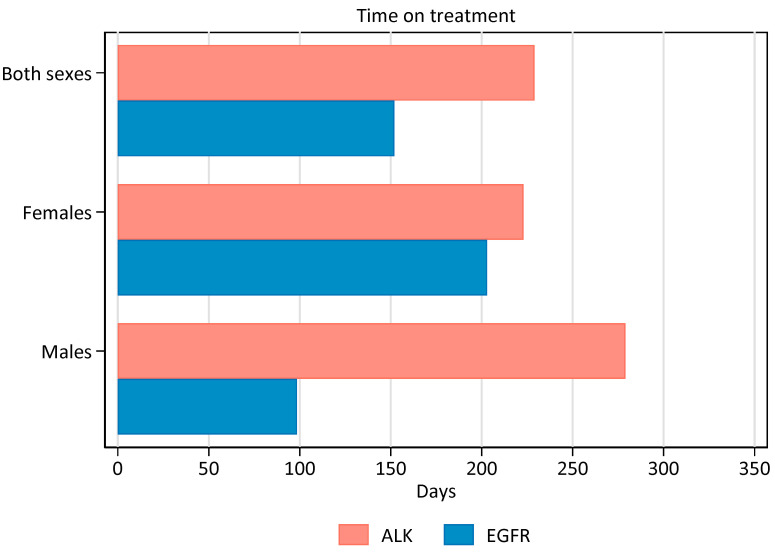
Median number on days on TKI treatment for the period 2016–2020. Patients dispensed only once are calculated with zero days on treatment.

**Figure 6 cancers-15-01505-f006:**
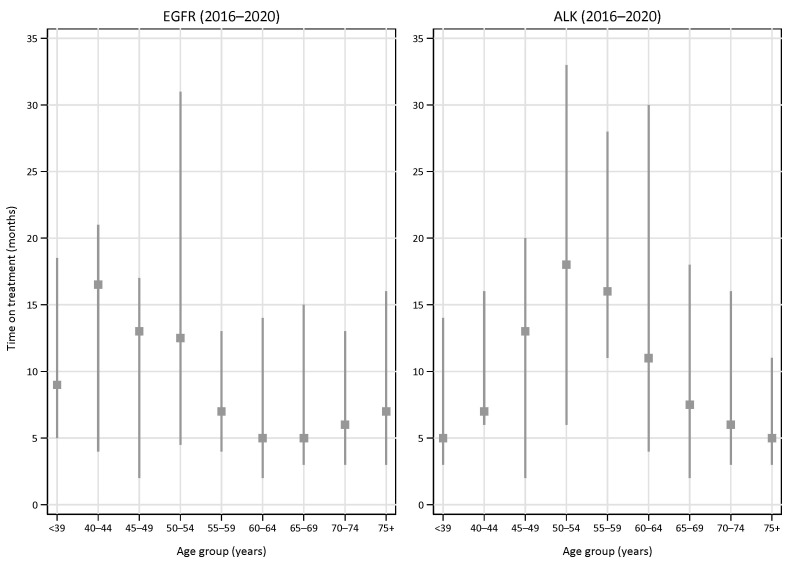
Median time on treatment (months)with EGFR-TKI (left panel) or ALK-TKI (right panel) in various age groups in the period 2016–2020. Error bars indicate interquartile ranges.

**Figure 7 cancers-15-01505-f007:**
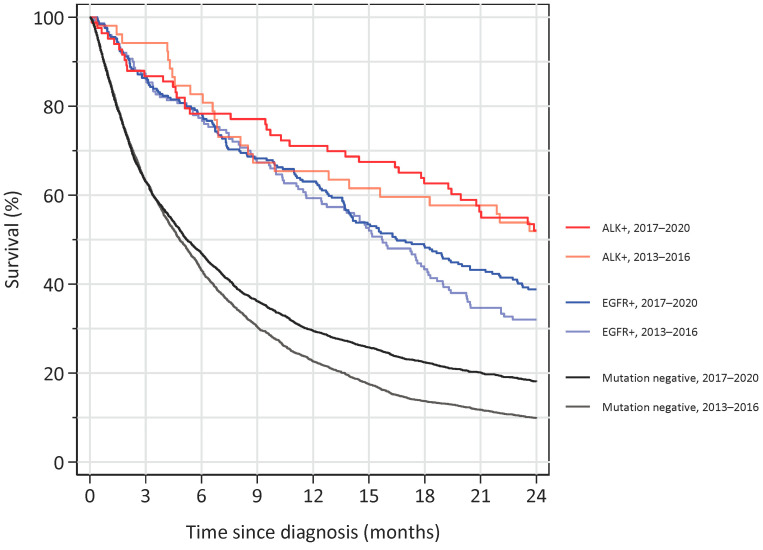
Overall survival for non-squamous NSCLC diagnosed at stage IV from either 2013–2016 or from 2017–2020 and followed until June 2022. Patients diagnosed with ALK mutations are shown in red colours, EGFR mutations are shown in blue colours, and patients without ALK- or EGFR-mutations are shown in a black/grey colour.

## Data Availability

Data are available from Cancer Registry of Norway and the Norwegian Prescription Database.

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
