# Peer review of "Real-World Data on EGFR and ALK Testing and TKI Usage in Norway—A Nation-Wide Population Study"

_cancers, 2023, doi:10.3390/cancers15051505_

Round 1

Reviewer 1 Report

Real world evidence and nation-wide data are extremely useful ancillary to the data generated in randomized clinical trials.

I do have couple of comments in regard to the manuscript submitted.

1.     Reflex testing has been implemented in Norway, eg patients with all stages of NSCLC are tested. Hence, it becomes important to differentiate patients who are eligible for TKI treatment.

The Authors state that: ‘Around 60% and 70% of EGFR- and ALK-positive patients were treated with TKIs, respectively.’

This information has little value when stage or palliative setting is unknown, clinically relevant question here what is the proportion of patients who were EGFR/ALK positive, had advanced disease and were eligible for TKI and actually started treatment (who and why some patients do not start treatment).

On these grounds, Figure 6 is misleading, indicated high rate of patients in whom there is no indication for treatment, and could be deleted. The Authors could also discuss whether information on lung cancer stage is available.

2.     I suggest the Authors clearly state in summary/abstract/conclusions how their real world data compare with the data from randomized trials, eg using time on treatment as surrogate for PFS. 

Instead of presenting data on packages used, time on treatment is recommended, as this facilitates reading.

3.     The Authors state that ‘….real-world replication of findings in clinical trials indicate that … (Abstract, page 1).

The Authors also state that ‘median duration for EGFR-TKI usage was 152 days (99 for males and 203 for females), (3.3 Time on treatment, page 8). However, this time is significantly shorter (median PFS 10 months or longer) than reported in randomized trials.

This discrepancy should be discussed and presented in line with results.

4.     To make this manuscript even stronger, the Authors could make attempt to retrieve overall survival data from cancer registry or at least discuss why this has not been done.

The Authors state that ‘Information on histology subtypes, EGFR- and ALK-test activity comes from the Cancer Registry of Norway (CRN).’ It is highly likely that CRN also has date and cause of death.

5.     Instead of using ATC codes, the Authors should present drug names by active ingredient. Reader is not familiar what drug is behind each ATC code (2. Material and methods, page 2).

6.     Treatment landscape of EGFR and ALK positive patients has changed with implementation of more effective next generation TKIs. I suggest the Authors present time on first line treatment by 1st/2ndgeneration EGFR TKI versus osimertiniib, and by crizotinib versus other ALK inhibitors. OS could be added if available.

7.     Impression is that too much emphasizes has been placed on age and gender comparison. For instance, Figure 4 and 5 add very little value to the manuscript.

Author Response

We thank this reviewer for insightful comments, and we believe the manuscript is much improved by the suggestions which to a high degree have been followed. Please see below for a point-to-point response (our responses in italics), and also the uploaded revised manuscript with "track changes" for specific changes made.

Reviewer 1

Real world evidence and nation-wide data are extremely useful ancillary to the data generated in randomized clinical trials.

I do have couple of comments in regard to the manuscript submitted.

  1. Reflex testing has been implemented in Norway, eg patients with all stages of NSCLC are tested. Hence, it becomes important to differentiate patients who are eligible for TKI treatment.

The Authors state that: ‘Around 60% and 70% of EGFR- and ALK-positive patients were treated with TKIs, respectively.’

This information has little value when stage or palliative setting is unknown, clinically relevant question here what is the proportion of patients who were EGFR/ALK positive, had advanced disease and were eligible for TKI and actually started treatment (who and why some patients do not start treatment).

On these grounds, Figure 6 is misleading, indicated high rate of patients in whom there is no indication for treatment, and could be deleted. The Authors could also discuss whether information on lung cancer stage is available.

We agree with the reviewer that figure 6 can be perceived misleading on the grounds described, and have therefore deleted the figure. We have extended the text in Results and the aspects raised by the reviewer are elucidated in the Discussion.

  1. I suggest the Authors clearly state in summary/abstract/conclusions how their real world data compare with the data from randomized trials, eg using time on treatment as surrogate for PFS. 

Instead of presenting data on packages used, time on treatment is recommended, as this facilitates reading.

We have adjusted the Abstract and Conclusion according to the suggestions raised by the reviewer, and also added overall survival figures as described in point 6 below. For the time on treatment, we have converted number of packages to months on treatment, as we agree with the reviewer that this facilitates reading.

  1. The Authors state that ‘….real-world replication of findings in clinical trials indicate that … (Abstract, page 1).

The Authors also state that ‘median duration for EGFR-TKI usage was 152 days (99 for males and 203 for females), (3.3 Time on treatment, page 8). However, this time is significantly shorter (median PFS 10 months or longer) than reported in randomized trials.

This discrepancy should be discussed and presented in line with results.

We have extended the discussion of this fact, with several possible reasons for the discrepancy, in the Discussion section.

  1. To make this manuscript even stronger, the Authors could make attempt to retrieve overall survival data from cancer registry or at least discuss why this has not been done.

The Authors state that ‘Information on histology subtypes, EGFR- and ALK-test activity comes from the Cancer Registry of Norway (CRN).’ It is highly likely that CRN also has date and cause of death.

We have added survival data for ALK- and EGFR-mutated patients as well as non-mutated patients (see also point 6 below).

  1. Instead of using ATC codes, the Authors should present drug names by active ingredient. Reader is not familiar what drug is behind each ATC code (2. Material and methods, page 2).

We have added generic drug names to the ATC codes.

  1. Treatment landscape of EGFR and ALK positive patients has changed with implementation of more effective next generation TKIs. I suggest the Authors present time on first line treatment by 1st/2ndgeneration EGFR TKI versus osimertinib, and by crizotinib versus other ALK inhibitors. OS could be added if available.

Unfortunately, we do not have access to detailed information on time on first generation vs later generation drugs. However, we have added survival data for ALK- and EGFR-mutated patients as well as non-mutated patients, and divided this into two time periods (diagnosed in 2013-2016 vs 2017-2020), as we believe this will serve the purposes suggested by the reviewer, and also strengthen the overall presentation. The survival data are displayed in a new figure, labeled Figure 7), and also related to in Discussion section.

  1. Impression is that too much emphasizes has been placed on age and gender comparison. For instance, Figure 4 and 5 add very little value to the manuscript.

We agree that figure 4 can be omitted, but we believe figure 5 bears some important information on the age differences both between sexes, but foremost between ALK- and EGFR-mutated patients – so we suggest keeping this figure (now re-labeled to Figure 4). We have also reworded some of the age and gender comparisons.

Reviewer 2 Report

I liked this paper - comparing data from trial populations with real-world data is a critical part of outcomes research. This study used a national database for diagnosis, pathology and prescribing - which is as good as can be achieved.

The data is anonymous and retrospective and acts as a benchmarking audit - so there are no difficult ethical issues involved in this project.

The results act as a benchmark for audits in othetr healthcare systems - which is why I am very keen to see this published.

The discussion and comments were well balanced - and I hope the falling trend in ALK+ patients over time will be explored further in subsequent repeats of this project in the years ahead.

The data, tables, charts and presentation was excellent. It was easy to read and the inclusion of raw data meant that others can process the data as well.

These papers may not seem at first as glamorous as randomised trials - but the recording of such real world data, and open acess to the results and methodology is critical for all health systems.

With the strength of this data - I would love to see if a cost effectiveness audit could be performed to see if the programme matches the health economic predictions from the trials and in other health systems. This is irrelevant to the current publication - but the comprehensiveness of the data from Norway could make it one of the most useful studies in this area. The methodology could be similar to the French audit - Loubière S, Drezet A, Beau-Faller M, Moro-Sibilot D, et al; French Cooperative Thoracic Intergroup (IFCT). Cost-effectiveness of KRAS, EGFR and ALK testing for decision making in advanced nonsmall cell lung carcinoma: the French IFCT-PREDICT.amm study. Eur Respir J. 2018 Mar 15;51(3):1701467. doi: 10.1183/13993003.01467-2017. PMID: 29545318.

To the researchers and authors - many thanks for your hard work - Publication Recommended without changes

Author Response

We thank this reviewer for the kind feed-back, and the recommendation to publish without changes.

Reviewer 2

I liked this paper - comparing data from trial populations with real-world data is a critical part of outcomes research. This study used a national database for diagnosis, pathology and prescribing - which is as good as can be achieved.

The data is anonymous and retrospective and acts as a benchmarking audit - so there are no difficult ethical issues involved in this project.

The results act as a benchmark for audits in othetr healthcare systems - which is why I am very keen to see this published.

The discussion and comments were well balanced - and I hope the falling trend in ALK+ patients over time will be explored further in subsequent repeats of this project in the years ahead.

The data, tables, charts and presentation was excellent. It was easy to read and the inclusion of raw data meant that others can process the data as well.

These papers may not seem at first as glamorous as randomised trials - but the recording of such real world data, and open acess to the results and methodology is critical for all health systems.

With the strength of this data - I would love to see if a cost effectiveness audit could be performed to see if the programme matches the health economic predictions from the trials and in other health systems. This is irrelevant to the current publication - but the comprehensiveness of the data from Norway could make it one of the most useful studies in this area. The methodology could be similar to the French audit - Loubière S, Drezet A, Beau-Faller M, Moro-Sibilot D, et al; French Cooperative Thoracic Intergroup (IFCT). Cost-effectiveness of KRAS, EGFR and ALK testing for decision making in advanced nonsmall cell lung carcinoma: the French IFCT-PREDICT.amm study. Eur Respir J. 2018 Mar 15;51(3):1701467. doi: 10.1183/13993003.01467-2017. PMID: 29545318.

To the researchers and authors - many thanks for your hard work - Publication Recommended without changes

Reviewer 3 Report

This paper is clearly written and well organized. The introduction and background are reasonable given the premise of the paper. The authors explain very well in this paper about EGFR- and ALK-testing and TKI-usage in Norway a nationwide population study. I am recommending to publishing this paper without any modification.

Author Response

We thank this reviewer for the kind feed-back, and the recommendation to publish without any modification.

Reviewer 3

This paper is clearly written and well organized. The introduction and background are reasonable given the premise of the paper. The authors explain very well in this paper about EGFR- and ALK-testing and TKI-usage in Norway a nationwide population study. I am recommending to publishing this paper without any modification.

Round 2

Reviewer 1 Report

Points raised have been adequatelly addressed.